# Immunogenicity and Antiviral Response of Therapeutic Hepatitis B Vaccination in a Mouse Model of HBeAg-Negative, Persistent HBV Infection

**DOI:** 10.3390/vaccines9080841

**Published:** 2021-07-31

**Authors:** Anna D. Kosinska, Julia Festag, Martin Mück-Häusl, Marvin M. Festag, Theresa Asen, Ulrike Protzer

**Affiliations:** 1Institute of Virology, Technical University of Munich/Helmholtz Zentrum München, D-81675 Munich, Germany; anna.kosinska@tum.de (A.D.K.); julia.festag@gmx.de (J.F.); martin.mueck-haeusl@tum.de (M.M.-H.); marvin.festag@gmail.com (M.M.F.); theresa.asen@tum.de (T.A.); 2German Center for Infection Research (DZIF), Munich Partner Site, D-81675 Munich, Germany

**Keywords:** hepatitis B virus, chronic hepatitis B, therapeutic vaccination, prime/boost vaccination, hepatitis B virus e antigen, pre-core mutation

## Abstract

During the natural course of chronic hepatitis B virus (HBV) infection, the hepatitis B e antigen (HBeAg) is typically lost, while the direct transmission of HBeAg-negative HBV may result in fulminant hepatitis B. While the induction of HBV-specific immune responses by therapeutic vaccination is a promising, novel treatment option for chronic hepatitis B, it remains unclear whether a loss of HBeAg may influence its efficacy or tolerability. We therefore generated an adeno-associated virus (AAV)-vector that carries a 1.3-fold overlength HBV genome with a typical stop-codon mutation in the pre-core region and initiates the replication of HBeAg(−) HBV in mouse livers. Infection of C57BL/6 mice established persistent HBeAg(−) HBV-replication without any detectable anti-HBV immunity or liver damage. HBV-carrier mice were immunized with *TherVacB*, a therapeutic hepatitis B vaccine that uses a particulate HBV S and a core protein for prime vaccination, and a modified vaccinia Ankara (MVA) for boost vaccination. The *TherVacB* immunization of HBeAg(+) and HBeAg(−) HBV carrier mice resulted in the effective induction of HBV-specific antibodies and the loss of HBsAg but only mild liver damage. Intrahepatic, HBV-specific CD8 T cells induced in HBeAg(−) mice expressed more IFNγ but showed similar cytolytic activity. This indicates that the loss of HBeAg improves the performance of therapeutic vaccination by enhancing non-cytolytic effector functions.

## 1. Introduction

Chronic HBV infection is the leading cause of liver cirrhosis and hepatocellular carcinoma (HCC), resulting in 880.000 deaths annually [1]. Recent reports estimate that approximately 3.5% of the world population suffers from chronic hepatitis B (CHB) [1]. Spontaneous resolution of the infection occurs only in less than 1% of these individuals [2]. In addition, treatment with available antivirals is rarely curative [3]. This poses an urgent medical need for the development of novel therapeutic strategies against CHB.

Increasing evidence suggests that immunotherapeutic approaches aiming to induce long-term HBV-specific immune control and the elimination of infected hepatocytes might achieve a hepatitis B cure (reviewed in [4]). This notion is supported by the fact that effective antiviral B cell, CD4, and CD8 T cell responses are capable of clearing HBV during acute self-limiting infection [5,6]. In contrast, HBV persistence correlates to poor B- and T cell responses; CD4+ and CD8+ T cells are scarce and partially dysfunctional [7,8,9], thereby restoring HBV-specific immunity by means of a potent and rationally designed therapeutic vaccine (reviewed in [10,11]) might be an effective treatment option for CHB patients.

We have previously developed a therapeutic hepatitis B vaccine, termed *TherVacB*, which follows a heterologous prime–boost regimen [12]. There are two injections of adjuvanted, particulate, recombinant HBV S (HBsAg) and core antigens (HBcAg) that are used to activate CD4 T cells, to induce neutralizing antibodies that capture circulating HBsAg, to prevent virus spread, and to simultaneously prime CD8 T cell responses. This is followed by one injection of an MVA vector expressing HBV antigens to selectively boost HBV-specific T cells [12]. In preclinical mouse models of persistent infection with low-to-intermediate HBV levels, *TherVacB* elicited strong HBV-specific immune responses, which resulted in the long-term immune control of HBV [12,13]. Nevertheless, high HBV antigen expression levels in the liver negatively influence vaccine-induced CD8 T cell response and interfere with the antiviral effect [12,13]. Whether and to which extent circulating HBsAg and HBeAg contribute to the low *TherVacB* efficacy remains unclear.

Mouse studies underline the role of HBeAg in the establishment of persistent HBV infection and the induction of virus-specific T cell tolerance [14,15]. In contrast to HBsAg, HBeAg becomes undetectable over-time during the course of CHB in a high proportion of patients [16]. The seroconversion of HBeAg to the corresponding anti-HBe antibodies is preceded by the reactivation of the host immune responses and is associated with a decrease in HBV DNA levels, liver disease remission, and a lower HCC incidence [17]. However, some HBeAg-negative chronic carriers may maintain high-level HBV replication and may show active liver disease [17]. This correlates to the emergence of the HBeAg(−) strains of HBV, which harbour mutations in the pre-core region and/or the core promoter of viral genome [18,19]. Previous reports suggest that alterations in the HBV pre-core/core region might result in the development of fulminant and severe hepatitis upon new infection [20,21] or might affect the response to antiviral treatment [22,23,24]. This indicates that HBeAg-negative HBV may activate stronger immune responses. However, how HBeAg modulates immune responses and the efficacy of therapeutic vaccination has not been systematically investigated. According to current knowledge, we have reasoned that a lack of HBeAg expression might result in an enhanced vaccine-mediated immunogenicity and antiviral effects. Nevertheless, this raises the concern whether inducing highly effective HBV-specific CD8 T cells via therapeutic vaccination might enhance liver damage and might result in liver failure in HBeAg(−) hepatitis B patients. In this study we therefore investigated how HBeAg status influences *TherVacB*-mediated HBV-specific immune responses and the elimination of persistent HBV-infection using a novel AAV-HBV mouse model.

## 2. Materials and Methods

### 2.1. Generation and Production of AAV-HBV WT and AAV-HBV-E(−)

AAV transfer plasmid encoding for the WT HBV sequence, pAAV-HBV1.3WT, was generated by inserting 1.3 copies of genotype D of the HBV genome (GenBank accession number: V01460.1) amplified by the pT-HBV1.3WT plasmid into an AAV2 ITR containing plasmid using the HindIII and SacI restriction enzyme site. To obtain the HBeAg(−) variant of pAAV-HBV1.3-e(−), typical point mutation G1896A in the pre-core region of the HBV genome was introduced by a polymerase chain reaction (PCR)-based site directed mutagenesis. This mutation generates a stop-codon in the E protein open reading frame and abolishes HBeAg expression [19].

The AAV-HBV and AAV-HBV-e(−) vector genomes of Serotype 2 were packaged in an AAV Serotype 8 capsid and produced using a standard triple transfection as described [25]. Briefly, HEK239T cells were transfected with the HBV-encoding AAV transfer plasmid (pAAV-HBV1.3-WT or pAAV-HBV1.3-e(−)), the AAV packing plasmid, and the adenovirus helper plasmid (pXX6-80 and pXR8) using linear polyethylenimine (Polysciences, Warrington, PA). After 72 h of transfection, cells were lysed by three times through a freeze–thaw cycle (−80 °C/37 °C each for 15 min) in lysis buffer (50 mM Tris-HCl (pH7.5)), 150 mM NaCl, and 5 mM MgCl2). The cell lysate was incubated with benzonase (Sigma, St. Louis, MI) at 37 °C for 30 min and was subjected to iodixanol gradient ultracentrifugation afterwards. The recombinant AAVs were recovered from the 40% iodixanol layer and titrated by means of quantitative PCR using an AAVpro Titration Kit Ver.2 (Takara Bio, Kusatsu, Japan) according to the manufacturer’s protocols.

### 2.2. Ethical Statement

Animal experiments were conducted in strict accordance with the regulations of the German Society for Laboratory Animal Science (GV-SOLAS) and the health laws of the Federation of European Laboratory Animal Science Associations (FELASA). Experiments were approved by the District Government of Upper Bavaria (permission numbers: 55.2-1-54-2532-120-12 and 55.2-1-54-2532-112-13). Following 3R principles, each in vivo experiment was performed once. Mice were kept in a specific pathogen-free facility under appropriate biosafety levels, following institutional guidelines.

### 2.3. AAV-HBV Transduction

Wild-type C57BL/6 mice (haplotype H-2^b/b^) were purchased from Charles River Laboratories (Schulzfeld, Germany). Persistent HBV replication in wild-type mice was established by the intravenous injection of either 3 × 10^10^ or 1 × 10^10^ genome equivalents (geq) of AAV-HBV WT and AAV-HBV ΔE.

### 2.4. Therapeutic Hepatitis B Vaccination

Mice were immunized with a protein prime followed by a recombinant modified vaccinia Ankara (MVA) vector boost vaccination scheme, as described previously [12,13]. Mice were immunized intramuscularly into the quadriceps muscles of both hind limbs while under isoflurane anaesthesia. Protein immunization with 10 µg of particulate HBsAg (genotype A, adw) and HBV core protein (genotype D, ayw, kindly provided by APP Latvijas Biomedicinas, Riga, Latvia) adjuvanted with 10 µg cyclic di-adenylate monophosphate (c-di-AMP) (InvivoGen, San Diego, CA, USA) was administered twice at a 2-week interval. After a period of two weeks after the second protein vaccination, mice received 3 × 10^7^ infectious units each of recombinant MVA vectors expressing HBV S or the HBV core protein (both genotype D, ayw).

### 2.5. Isolation of Lymphocytes from Spleen and Liver

Splenocytes were isolated as described previously [12]. Liver-associated lymphocytes were isolated and were purified by density gradient centrifugation, as described in [12]. Briefly, mouse liver was perfused with pre-warmed PBS and forced through a 100 µm nylon cell strainer (BD Falcon, Franklin Lakes, NJ, USA). Cell pellets were suspended in 10 mL of prewarmed enzyme solution containing 1 mg/mL of collagenase type IV (Worthington, Lakewood, NJ, USA) in RPMI 1640 medium supplemented with 10% fetal bovine serum (Gibco, Thermo Fischer Scientific, Darmstadt, Germany) and were digested for 30 min at 37 °C. Cell pellets were then resuspended in 40% Percoll (GE Healthcare, Munich, Germany) and were layered on 80% Percoll solution and centrifuged at 1600× *g* for 20 min without brakes for density separation.

### 2.6. Detection of HBV-Specific CD8 T Cells by Multimer and Intracellular Cytokine Staining

Core-specific CD8 T cells were detected through staining with MHC class I multimers conjugated with the H-2K^b^-restricted HBV core-derived peptide C_93–100_ (C_93_, MGLKFRQL), as described previously [13,26]. As a control, staining with multimer conjugated with the ovalbumin-derived peptide S8L_257_ (OVA_S8L_, SIINFEKL) was performed. Prior to use, C_93_ and OVA_S8L_ multimers (kindly provided by Dirk Busch, Technical University of Munich, Germany) were labeled with Streptactin-PE (IBA Lifesciences, Göttingen, Germany), as previously described [13,26].

For intracellular cytokine staining, cells were stimulated for 16h in the presence of 1 mg/mL Brefeldin A (Sigma-Aldrich, Taufkirchen, Germany) with the synthetic core-derived peptide C_93_, the S-derived peptide S_190-197_ (S_190_, VWLSAIWM), or the control ovalbumin-derived peptide (OVA_S8L_) added to a final concentration of 1 µg/mL. Cell surface staining was performed using the anti-CD8 (clone 56.6–7; BD Biosciences, Heidelberg, Germany) and the anti-CD4 (clone L3T4; BD Biosciences) T cell antibodies. Dead cells were excluded from analysis by Fixable Viability Dye eF780 (eBioscience, Frankfurt, Germany) staining. Intracellular cytokine staining was performed using a Cytofix/Cytoperm Kit (BD Biosciences, Heidelberg, Germany) according to the manufacturer’s instructions with the anti-IFNγ (clone XMG1.2; eBioscience) and the cross-reactive anti-human granzyme B (clone: GRB04; Invitrogen, Carlsbad, CA) antibodies. Data were acquired on a CytoflexS (Beckmann Coulter) flow cytometer. Analyses were performed using FlowJo-Version10 software (Tree Star, Ashland, OR).

### 2.7. Serological and vVirological Analyses

HBsAg and HBeAg levels were quantified on an Architect platform using the quantitative HBsAg test (Ref.: 6C36-44) and the HBeAg reagent kit (Ref.: 6C32-27) with HBeAg quantitative calibrators (Ref.: 7P24-01). Anti-HBs and anti-HBe antibodies were determined using anti-HBs (Ref.: 7C18-27) or anti-HBe (Ref.: 6C34-25) tests for an Architect platform (all: Abbott Laboratories, Wiesbaden, Germany). Anti-HBc were detected using an Enzygnost anti-HBc monoclonal test (Siemens Healthcare Diagnostics, Erlangen, Germany). ALT activity was measured in serum samples diluted 1:4 with PBS using a Reflotron GPT/ALT test (Roche, Mannheim, Germany).

DNA was extracted from 100µL mouse serum or 20 mg of liver tissue using a NucleoSpin Tissue DNA Kit (Macherey-Nagel, Dueren, Germany) according to the manufacturer’s instructions.

Quantitative PCR for HBV DNA in mouse serum was performed on an Applied Biosystems 7500 Real time PCR system using the primers HBV-1464-Fw: 5′-GGACCCCTTCTCGTGTTACA-3′, HBV-1599-Rev: 5′- ACTGCGAATTTTGGCCAAGA-3′, and the HBV-specific Taqman probe 5′-CTAGACTCGTGGTGGACTTCTCTCAATTTTCT-3′ (lower limit of quantification: 150 copies/mL serum). The amplification conditions were: 50 °C for 120 s and 95 °C for 600 s followed by 45 × (95 °C for 15 s, 60 °C for 60 s).

The quantification of intrahepatic HBV DNA was performed through real-time PCR with SYBR green according to the manufacturer’s instructions using a LightCycler 480 PCR system (both: Roche, Mannheim, Germany) and following the primers HBV-1745-Fw: 5′-GGAGGGATACATAGAGGTTCCTTGA-3′ and HBV-1844-Rev: 5′-GTTGCCCGTTTGTCCTCTAATTC-3′ (lower limit of quantification: approximately 10-100 copies/100 ng of liver tissue). The results were normalized to a single copy prion protein (PrP) gene (primers: PrP-Fw 5′-TGCTGGGAAGTGCCATGAG-3′ and PrP-Rev 5′- CGGTGCATGTTTTCACGATAGTA-3′). The amplification conditions for both of the PCR reactions were 95 °C for 300 s followed by 45 × (95 °C for 15 s, 60 °C for 10 s, and 72 °C for 25 s).

### 2.8. Immunohistochemistry

Liver tissue samples were fixed in 4% buffered formalin for 48 h and were paraffin embedded. Liver sections that were2-μm-thin were then prepared with a rotary microtome (HM355S, ThermoFisher Scientific, Waltham, USA). Immunohistochemistry was performed using a Bond Max system (Leica, Wetzlar, Germany, all reagents from Leica) with the anti-HBcAg primary antibody (Diagnostic Biosystems, Pleasanton, CA; 1:50 dilution) and a horseradish peroxide coupled secondary antibody. Briefly, the slides were deparaffinized using deparaffinization solution pre-treated with epitope retrieval solution (corresponding to citrate buffer pH6) for 20 min. Antibody binding was detected with a polymer refine detection kit without post primary reagent and was visualized with DAB as a dark brown precipitate. Counterstaining was done with haematoxylin. Slides were scanned using a SCN 400 slide scanner (Leica Biosystems, Nussloch, Germany), and HBcAg-positive hepatocytes were determined based on the localization, intensity, and distribution of the signal in 10 random view fields (40× magnification). The mean numbers of the HBcAg-positive hepatocytes were quantified per mm2.

### 2.9. Statistical Analyses

Statistical analyses were performed using GraphPad Prism version 5 (GraphPad Software Inc., San Diego, CA, USA). Statistical differences were analyzed using the Kruskal–Wallis test with Dunn’s multiple comparison correction, 2-way ANOVA with Bonferroni multiple comparison correction, and the Mann–Whitney test. *p*-values < 0.05 were considered significant.

## 3. Results

### 3.1. Generation and In Vivo Characterization of AAV Vectors Encoding 1.3 Overlength WT HBV Genome and Corresponding HBeAg(−) Variant

To establish a mouse model that allows for the persistent replication or HBV that lacks the expression of HBeAg, we first generated AAV-HBV vector encoding for a wild-type 1.3-overlength of genotype D of the HBV genome (WT). Next, the dominant pre-core stop-codon mutation at position 1896 (G to A), which abolishes the expression of HBeAg, was inserted into the 1.3-fold HBV genome in both the 5′ and 3′ ORFs [19] to obtain AAV-HBV-e(−).

To verify whether the new AAV-HBV1.3 vectors were capable of establishing persistent HBV replication *in vivo*, we infected immunocompetent C57BL/6 mice with equal doses of WT or e-minus variants. We monitored HBV parameters in the serum of the mice for 12 weeks after infection. After AAV-HBV-e(−) transduction, no HBeAg was detected in the sera of the mice (Figure 1A), confirming the functionality of HBeAg knock-out. In contrast, high serum HBeAg levels were detected in mice receiving AAV-HBV WT and remained stable over the monitoring period.

Both groups of mice demonstrated long-term stable HBsAg levels in the serum (Figure 1B), indicating the establishment of persistent HBV replication in the liver. Of note, serum HBsAg levels detected for HBeAg(−) mice were on average 0.5 log_10_ lower than the levels detected in the mice infected with the WT HBV genome, although these differences were not statistically significant. Serum alanine transaminase values (sALT) remained normal at all of the examined time points (Figure 1C), suggesting that no immune-mediated liver damage was neither induced upon infection with WT AAV-HBV nor with HBeAg(−) AAV-HBV-e(−). Consistently, no antibodies directed against HBsAg (anti-HBs; Figure 1D), the HBV core protein (anti-HBc; Figure 1E), or HBeAg (anti-HBe; Figure 1F) could be detected up to 12 weeks after AAV-HBV infection.

At week 12, we sacrificed the mice to investigate whether the G1896A stop codon mutation introduced in the pre-core region of AAV-HBV-e(−) could interfere with HBV replication or expression of core protein in the liver. The quantification of intrahepatic HBV DNA demonstrated no significant differences between the mice infected with either AAV-HBV WT or AAV-HBV-e(−) (Figure 1G). Immunohistochemistry staining of the liver tissue displayed a weaker core-specific signal in some HBV-positive hepatocytes of mice replicating HBeAg(−) HBV compared to WT HBV (Figure 1H). However, the quantification of the core-positive hepatocytes demonstrated only minor differences in their numbers between the two groups of mice (Figure 1I).

These results demonstrate that the AAV-mediated transfer of HBV genomes into the mouse liver allows the establishment of the persistent replication of WT and HBeAg(−) HBV in immunocompetent C57BL/6 mice. The minor differences in serum HBsAg and intrahepatic HBV core expression we observed maybe attributed to differences in the quality of the AAV-HBV vector preparation. Importantly, the transduction of mouse livers with HBeAg(−) HBV did neither activate strong anti-HBV immunity nor cause liver damage or the fulminant hepatitis, indicating that cofounding factors are required to explain the observation of fulminant and acute HBeAg-negative hepatitis B in humans.

### 3.2. TherVacB-Mediated Immunogenicity and Antiviral Efficacy in AAV-HBV Mice Deficient for HBeAg

Having established a suitable mouse model, we wanted to investigate the effect of *TherVacB* in HBeAg-negative, chronic HBV infection, which is observed in most patients eligible for therapeutic vaccination. To address the concern that *TherVacB* might induce liver damage in HBeAg(−) HBV carriers and to mimic a clinical situation, we aimed to establish HBV replication with low viremia and relatively low-level antigen expression, as this was expected to facilitate the induction of strong effector CD8 T cell responses [13].

After two weeks of AAV-HBV infection, we confirmed comparable but low-level HBV replication by determining HBsAg levels in serum of mice infected with AAV-HBV and AAV-HBV-e(−) (Figure 2A). As expected, in mice infected with AAV-HBV-e(−), serum HBeAg remained negative, whereas all mice who received WT AAV-HBV were positive for HBeAg (Figure 2B).

At four weeks post-infection, *TherVacB* immunization began (week 0). Mice received two protein immunizations with adjuvanted, particulated HBsAg and HBcAg at week 0 and 2, followed by a boost with recombinant MVA expressing HBV S and core proteins at week 4. End-point analyses were performed one week later, at the peak of the *TherVacB*-mediated effector immune responses. AAV-HBV or AAV-HBV-e(−) infected mice who were not vaccinated served as controls.

We compared the induction of HBV-specific B cells by determining antibody responses after *TherVacB* in mice replicating WT or HBeAg(−) HBV. *TherVacB* elicited equally high levels of anti-HBc (Figure 2C) and anti-HBs (Figure 2D) in HBeAg(−) and HBeAg(+) mice. A strong *TherVacB*-mediated anti-HBs response in immunized mice resulted in a rapid decrease of serum HBsAg, which dropped below the level of detection at week 4 (Figure 2E). Interestingly, the time kinetics of serum HBsAg clearance after the initiation of *TherVacB* observed in HBeAg(−) and HBeAg(+) mice was comparable, suggesting that HBeAg-negativity does not influence vaccine-mediated HBsAg to anti-HBs seroconversion. In the absence of vaccination, neither HBeAg(+) nor HBeAg(−) mice developed antibodies against HBV antigens (Figure 2C,D) and showed stable serum HBsAg levels throughout the experiment (Figure 2E). This confirms that although AAV-HBV infection resulted in a low-level of HBV infection, it could not be spontaneously resolved, even in the absence of HBeAg.

Shortly after the initiation of *TherVacB*, a peak in serum ALT levels was observed in mice replicating HBeAg(+) and HBeAg(−) HBV variants (Figure 3A), indicating that the vaccine induced effector CD8 T cell responses in the liver. There was no enhanced liver damage detected in mice infected with the HBeAg(−) mutant. The increase in the ALT mice demonstrated a tendency towards lower values in e(−) than in the WT AAV-HBV infected mice. In mice receiving WT AAV-HBV variant, a *TherVacB*-mediated increase in ALT levels correlated in time with a significant decrease of serum HBeAg levels (Figure 3B), which further confirmed the elimination of HBV-infected hepatocytes. Since this analysis was not possible in the AAV-HBV-e(−) mice, we have further investigated the antiviral efficacy of *TherVacB* through the quantification of HBV DNA in mouse serum and liver tissue. In both groups of mice, immunization with *TherVacB* efficiently reduced serum HBV DNA to levels below the limit of detection (Figure 3C). Moreover, intrahepatic HBV DNA significantly dropped in *TherVacB*-immunized mice compared to the corresponding unvaccinated controls, but most likely due to the fact that the remaining AAV-HBV DNA did not become negative (Figure 3D).

### 3.3. TherVacB Induces T Cell Responses in Mice Replicating HBeAg(+) and HBeAg(−) HBV

As the MVA vaccination aims to expand the effector T cell responses, we investigated whether HBV core-specific T cells were induced by *TherVacB* in mice replicating HBeAg(−) HBV, compared to those replicating WT HBV one week after MVA boost. Determining the frequencies of the total CD8 T cells in freshly isolated splenic and hepatic lymphocytes, we found a high number of CD8 T cells in the spleens of mice receiving *TherVacB* (Figure 4A). In addition, *TherVacB* led to an impressive infiltration of CD8 T cells into the liver, resulting in a 4-fold increase in CD8 T cells frequencies compared to the non-vaccinated mice (Figure 4A). These results were independent of HBeAg status since no significant differences were observed between the HBeAg(+) and HBeAg(−) mice.

To detect and characterize HBV-specific T cells, we used MHC multimers loaded with a peptide covering the HBV core amino-acid 93-100 (C_93_) that allows for the direct ex vivo staining of HBV-specific CD8 T cells without prior manipulation. Ex vivo C_93_-multimer staining of liver-infiltrating lymphocytes revealed that a significant amount of core-specific CD8 T cells was induced by vaccination, whereas only background responses were detected in the non-vaccinated animals (Figure 4B). Overall frequencies of C_93_-specific CD8 T cells after *TherVacB* were not different between the HBeAg(+) and HBeAg(−) mice (Figure 4C). In addition, no C_93_-specific CD8 T cells could be detected in the livers of mice who did not receive *TherVacB*, confirming that without vaccination, no HBV-specific CD8 T cell response was induced in the AAV-HBV mouse model.

We next investigated the expression of inhibitory programmed death receptor-1 (PD-1) on hepatic C_93_-specific CD8 T cells in the *TherVacB*-immunized mice since it was previously reported that PD-1 plays a key role in regulating the functionality of HBV-specific T cells in the liver [13]. We demonstrated that the fraction of PD-1-expressing C_93_-specific CD8 T cells was reduced in the livers of the mice replicating HBeAg(−), compared to those replicating HBeAg(+) HBV (Figure 4D). Moreover, the expression levels of PD-1 per cell were significantly lower in the C_93_-specific CD8 T cells isolated from the mice which were deficient for HBeAg (Figure 4D).

In splenocytes, which represent lymphocytes that circulate systemically, the proportion of C_93_-specific T cells was significantly lower than in the liver-associated lymphocyte population (Figure 4E). In addition, in splenocytes, the proportion of PD-1 expressing C_93_-specific T cells as well as the expression level was markedly reduced, and no difference between the HBeAg(+) and HBeAg(−) mice was detected (Figure 4F).

### 3.4. Vaccine-Induced T Cell Function Differs in Mice Replicating HBeAg(+) and HBeAg(−) HBV

Since functional HBV-specific CD8 T cells are crucial for the elimination of an established HBV infection [13,26], we next examined the functionality of *TherVacB*-induced CD8 T cell responses in the presence or absence of HBeAg.

We studied whether the high numbers of CD8 T cells elicited via therapeutic vaccination were HBV-specific and showed an antiviral effector function. For this purpose, we restimulated CD8 T cells from the liver or spleen ex vivo in the presence of the HBV core- and the S-derived peptides C_93_ and S_190_. In splenocytes, effector T cell responses indicated by core-specific IFNγ+ CD8 T cells were detected in all of the mice after *TherVacB* vaccination (Figure 5A), while circulating S-specific IFNγ+ CD8 T cells were barely detectable. In the liver, very high frequencies of HBV-specific, IFNγ+ CD8 T cell responses were detected, particularly in HBeAg(−) mice (Figure 5B,C). As was the case for circulating T cells, core-specific CD8 T cell responses were dominant in the liver, and significant S-specific CD8 T cell response were only detected in the HBeAg(−) mice (Figure 5B,C).

Because effector function is not only determined by cytokine secretion but also by cytotoxic T cell function, we also determined the secretion of granzyme B (GzmB). Frequencies of HBV core-specific GzmB+ CD8 T cells were significantly higher than background levels detected in the corresponding unvaccinated controls, whereas only low levels of S-specific IFNγ+ CD8 T cells were detected in the WT AAV-HBV group (Figure 5C,D). While the frequencies of IFNγ+ CD8 T cells were higher in the HBeAg(−) mice than in the HBeAg(+) mice, there was no difference in the frequencies of hepatic granzyme B-expressing core- and S-specific CD8 T cells between both groups of mice after *TherVacB* (Figure 5D). Taken together, the observed T cell phenotype suggested that the lack of HBeAg expression influences the functionality of HBV-specific T cells locally in the liver.

## 4. Discussion

In this study, we reported that therapeutic vaccination in the absence of secretory HBeAg is as effective as it is in HBeAg(+) HBV infection. The vaccination of HBeAg(−) mice with *TherVacB* resulted in the induction of strong HBV-specific antibody and CD8 T cell responses, which were comparable to those in the HBeAg(+) mice and led to a loss of HBsAg and intrahepatic HBV-DNA after a transient, moderate increase of ALT activity, indicating very limited liver damage. In the HBeAg(−) mice, interestingly, T cell functionality seemed improve because a higher expression of IFNγ was accompanied by lower PD-1 expression levels in hepatic HBV-specific CD8 T cells compared to HBeAg(+) infected mice. Importantly, the lack of HBeAg did not result in increased granzyme B expression, which could be attributed to the enhanced cytotoxicity of virus specific CD8 T cells. This demonstrated that the *TherVacB* is able to induce strong HBV-specific CD8 T cell responses in HBeAg(+) as well as HBeAg(−) HBV infection, leading to the significant elimination of infected hepatocytes. Consistently, the therapeutic vaccination primed effector HBV-specific CD8 T cell responses were equally strong in HBeAg(−) and HBeAg(+) mice and led to in comparable amounts of these cells in the liver. Interestingly, a difference in the functionality and the phenotype of the hepatic HBV-specific cells could be observed between the mice actively expressing HBeAg and mice deficient for HBeAg. Vaccine-induced HBV-specific CD8 T cells in HBeAg- mice produced more IFNγ and had lower PD-1 expression levels. However, no difference in the cytolytic functions of these cells was observed, as shown by comparable GzmB expression and serum ALT elevations. This means that while HBeAg(−) HBV carrier mice showed stronger hepatic IFNγ responses, this did not lead to enhanced liver damage, at least in the AAV-HBV mouse model. Independent of the HBeAg status, *TherVacB* induced a predominantly core-specific CD8 T cell response, which most likely are main players responsible for HBV clearance. This is supported by the notion that core-specific CD8 T cells have been proven to be associated with HBV control in chronic hepatitis B patients [9].

The loss of HBeAg expression during the natural course of chronic HBV infection might be associated with the development of mutations in the pre-core region [18,19]. In our study we have established an HBeAg(−) AAV-HBV mouse model through the insertion of the of the most commonly observed stop-codon mutation, G1896A, which abolishes HBeAg expression [18]. Our results clearly demonstrate that the infection of immunocompetent mice with an HBeAg(−)-HBV variant results in the establishment of long-term persistent replication, and neither the spontaneous development of HBV-specific immunity, nor any significant liver damage could be observed. Several studies demonstrated that infection with the G1896A HBV variant might result in fulminant hepatitis [20,21]. However, more recent findings have demonstrated that in chronic hepatitis B, additional mutations occurring in the basal core promoter are required to enhance HBV virulence and exacerbate liver disease [27]. The accumulation of mutations in the pre-core region and/or the core promoter of the HBV genome occurs naturally during the years of persistent HBV replication [28].

Although our results are certainly very interesting for the future clinical development of therapeutic vaccination, they have to be interpreted with some caution. In our model, HBV replication mainly depends on the HBV genome initially being transferred by the AAV-HBV vector and thus does not fully reflect the pre-core/core promoter mutation rates occurring in chronic hepatitis B patients. In addition, in the HBeAg(−) AAV-HBV mouse model, no development of anti-HBe or serum ALT flares associated with liver inflammation due to the activation of HBV-specific immune response was detected, which constitutes another important limitation of the model compared to HBeAg(−) chronic hepatitis B [17].

Triggering HBV-specific adaptive immune responses via therapeutic vaccination, which may achieve sustained control of virus replication and an HBV cure, represents a novel treatment option for chronic hepatitis B. The investigated G1896A HBeAg mutation has been shown to be associated with an improved response to interferon-alpha treatment in CHB patients [24], underlying the immunosuppressive role of HBeAg during HBV infection [14]. *TherVacB* vaccination of HBeAg(−) HBV carrier mice did not reveal a stronger antiviral effect compared to HBeAg(+) HBV replication. The therapeutic vaccination resulted in a comparable HBV-specific antibody response, HBsAg seroconversion, and the induction of HBV-specific CD8 T cell responses in HBeAg(−) and HBeAg+ mice, showing that HBeAg expression did not influence peripheral HBV-specific immune tolerance in our model. However, HBeAg expression influenced the functionality of the HBV-specific CD8 T cells in the liver, suggesting that the absence of HBeAg may improve the non-cytolytic activity of vaccine-elicited hepatic CD8 T cells.

HBV-specific CD8 T cells play a crucial role in the elimination of HBV infection [5,6], which could be achieved by non-cytolytic and cytolytic mechanisms [29]. The secretion of antiviral cytokines, especially IFNγ, targets the intrahepatic persistent HBV form of covalently closed circular DNA (cccDNA) in a non-cytolytic fashion [30]. As HBV is a non-cytopathic virus, CD8 T cell-mediated lysis of infected hepatocytes is associated with the elevation of serum ALT levels observed at different stages of HBV infection [17]. We have previously shown that effective HBV immune control mediated by therapeutic vaccination relies on both effector functions of antiviral CD8 T cells in preclinical mouse models [12,13,26]. Here, we demonstrated that *TherVacB* resulted in the induction of comparable HBV-specific CD8 T cell responses in HBeAg(−) and HBeAg(+) mice, as shown by T cell analyses in the spleen. The HBeAg status did not influence the infiltration of HBV-specific CD8 T cells into the liver, nor did it influence their early cytolytic activity, as confirmed by the detection of a moderate serum ALT peak shortly after the initiation of *TherVacB* in the WT and e(−) AAV-HBV-infected mice.

HBeAg is produced from the mRNA initiating 29 codons upstream of the HBV core protein and in frame with the core protein start codon [31]. Hence, HBeAg and the core protein share the same amino acid sequence. We reasoned that the stop-codon mutation, carrying by HBeAg(−) AAV-HBV variant, not only abolishes HBeAg secretion into the blood but also reduces the protein level expressed in the hepatocytes, thereby resulting in a lower level of antigen presentation of the CD8 T cell epitopes, which are identical between HBeAg and the core protein. We have previously demonstrated that the functionality of intrahepatic HBV-specific CD8 T cells strongly correlates to the liver antigen expression levels [13]. This suggests that stronger IFNγ secretion by core-specific CD8 T cells in HBeAg(−) mice can be contributed to the less frequent recognition of the C_93_-epitope in the liver, which might also explain the lower PD-1 expression levels.

The influence of HBeAg expression on increased IFNγ production by hepatic S-specific CD8 T cells is more difficult to address. It is known that the local liver microenvironment significantly shapes intrahepatic HBV-specific immunity [32]. We can only speculate that the secretion of antiviral cytokines and inflammation in the liver elicited by highly activated core-specific CD8 T cells could trigger stronger IFNγ expression in S-specific CD8 T cells.

There is increasing evidence that the coinhibitory receptor PD-1 is associated with T cell exhaustion and plays a central role inducing T cell tolerance in the liver and in down-regulating intrahepatic antiviral responses, e.g., responses to HBV [33,34]. The lower levels of PD-1 expression on HBV core-specific CD8 T cells in the livers of the vaccinated HBeAg(−) mice could be an additional explanation for the differences in IFNγ secretion. However, the fact that HBV infection was eliminated equally efficiently in both mouse models demonstrated that upon *TherVacB,* the intrahepatic HBV-specific CD8 T cells isolated from HBeAg(+) were still functional, indicating that PD-1 expression could also merely be a sign of T cell activation. This is in concurrence with the observation that increased PD-1 expression on HBV-specific CD8 T cells during acute HBV infection did not impair the resolution of the infection [35]. Moreover, recent reports have demonstrated high PD-1 expression on HBV-specific liver-resident memory T cells, which were capable of producing high levels of effector antiviral cytokines such as IFNγ and tumor necrosis factor (TNF) upon restimulation [36]. Our findings may thus indicate that the lack of HBeAg expression accelerated memory T cell formation due to the lower antigen load, which might explain the differences in PD-1 expression on core-specific CD8 T cells. Nevertheless, this issue requires more detailed immunological analyses.

## 5. Conclusions

Taken together, one the one hand, our study demonstrates that infection of fully immune competent mice with an HBeAg(−) strand of HBV does not induce spontaneous T cell responses that clear the virus or cause liver damage. On the other hand, it demonstrates that therapeutic vaccination in HBeAg(−) HBV carrier mice is safe and efficacious, induces strong B and T cell responses, and may even result in improved T cell functionality than in HBeAg(+) carriers. Our data encourage the use of *TherVacB* in clinical trials in HBeAg(−) chronic hepatitis B patients

## Figures and Tables

**Figure 1 vaccines-09-00841-f001:**
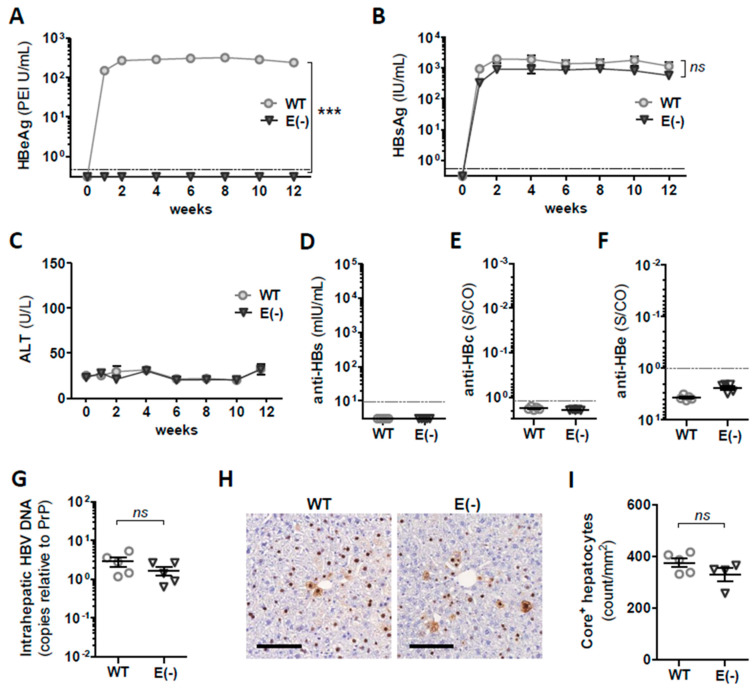
Establishment of persistent HBV replication in mice after infection with AAV-HBV encoding 1.3 overlength wild-type and HBeAg(−) HBV genomes. C57BL/6 mice (*n* = 5) were intravenously injected with 3 × 10^10^ GE of AAV-HBV vector carrying a 1.3-fold overlength the HBV genome (WT) or the HBeAg(−) variant AAV-HBV-e(−) (E(−)). At 12 weeks AAV-HBV infection, animals were sacrificed and were analyzed. Kinetics of (**A**) HBsAg, (**B**) HBeAg, and (**C**) sALT serum levels. (**D**) Anti-HBc, (**E**) anti-HBs, and (**F**) anti-HBe levels detected in mouse serum at week 12. (**G**) HBV DNA levels determined at week 12 in liver tissue lysates determined by quantitative PCR. (**H**) Representative immunohistochemistry of the HBV core and (**I**) the quantification of core-positive hepatocytes per mm2. Scale bar represents 100 µm. (**A**–**C**) Mean value of *n* = 5 mice per group ± standard error of mean, and in (**D**–**G**,**I**), individual mice are shown. (**A**,**B**,**D**–**F**) Dotted lines indicate the sensitivity cut-off of the assay. Statistical analyses were performed using (**A**,**B**) two-way ANOVA with Bonferroni multiple comparison correction and (**G**,**I**) the Mann–Whitney test. Asterisks mark statistically significant differences: *** *p* < 0.001; *ns*—not significant.

**Figure 2 vaccines-09-00841-f002:**
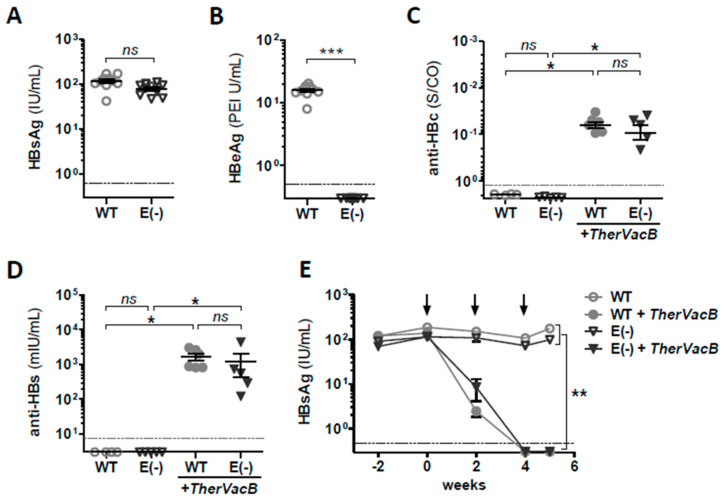
Induction of HBV-specific antibody response and HBsAg seroconversion after therapeutic vaccination in mice replicating HBeAg(+) and HBeAg(−) HBV. C57BL/6 mice (*n* ≥ 4) were intravenously injected with a low dose (1 × 10^10^ GE) of AAV-HBV vector carrying a 1.3-fold overlength HBV genome (WT) or the HBeAg(−) variant AAV-HBV-e(−) (E(−)). Levels of serum (**A**) HBsAg and (**B**) HBeAg detected 2 weeks post AAV-HBV transduction. Vaccination with *TherVacB* was initiated at week 4 after AAV-HBV infection (week 0). Immunization with adjuvanted, particulate HBsAg and HBcAg was performed at week 0 and 2. Boosting immunization with recombinant MVA expressing HBV S and core proteins was applied at week 4. Mice who received no vaccine served as controls. At week 5 after the start of *TherVacB* treatment, i.e., one week after MVA-boost, animals were sacrificed and analyzed. (**C**) Anti-HBc and (**D**) anti-HBs levels detected in mouse serum at week 5. (**E**) Kinetics of HBsAg serum levels; black arrows indicate the time points of *TherVacB* immunizations. (**A–D**) Symbols represent individual mice, and in (**E**), mean value of *n* ≥ 4 mice per group ± standard error of mean is shown. (**A**–**E**) Dotted lines indicate the sensitivity cut-off of the assay. Statistical analyses were performed using (**A**,**B**) the Mann–Whitney test, (**C**,**D**), the Kruskal–Wallis test with Dunn’s multiple comparison correction, and (**E**) two-way ANOVA with Bonferroni multiple comparison correction. Asterisks mark statistically significant differences: *** *p* < 0.001; ** *p* < 0.01; * *p* < 0.05; *ns*—not significant.

**Figure 3 vaccines-09-00841-f003:**
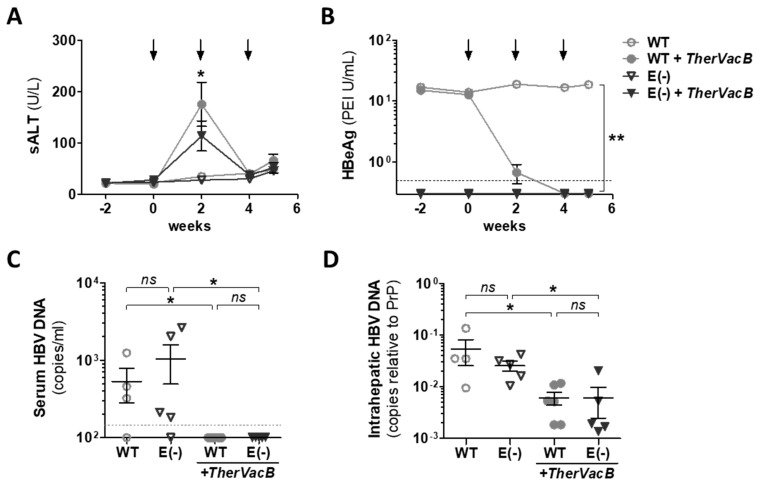
Antiviral effect of therapeutic vaccination in mice replicating HBeAg(+) and HBeAg(−) HBV. Persistent HBV replication was established in C57BL/6 mice using AAV-HBV encoding for a wild-type (WT) or AAV-HBV-e(−) encoding for an HBeAg(−)HBV variant (E(−)) (see Figure 2). Vaccination with *TherVacB* was initiated at week 4 after AAV-HBV infection (week 0). End-point analyses were performed at week 5, i.e., one week after MVA-boost. Kinetics of (**A**) ALT and (**B**) HBsAg levels in serum are given; black arrows indicate *TherVacB* immunizations. (**C**,**D**) HBV DNA levels were determined at week 5 in serum (**C**,**D**) liver tissue lysates through quantitative PCR. (**A**,**B**) Mean value of *n* ≥ 4 mice per group ± standard error of mean is shown and (**C**) symbols represent individual mice. Dotted lines in (**B**) indicate the sensitivity cut-off of the assay and (**C**) the lower quantification limit. Statistical analyses were performed using (**A**,**B**) two-way ANOVA with Bonferroni multiple comparison correction and (**C**,**D**) the Kruskal–Wallis test with Dunn’s multiple comparison correction. Asterisks mark statistically significant differences: ** *p* < 0.01; * *p* < 0.05; *ns*—not significant.

**Figure 4 vaccines-09-00841-f004:**
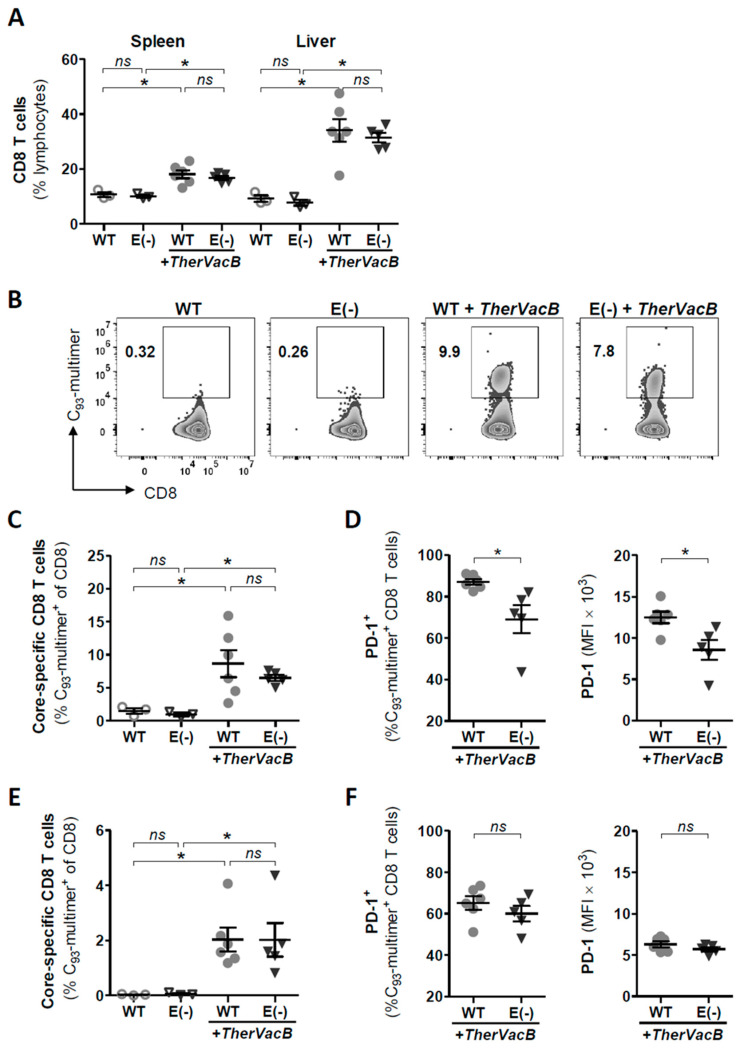
Phenotype of hepatic core-specific CD8 T cells after therapeutic vaccination in in AAV-HBV-infected HBeAg+ and HBeAg(−) mice. Persistent HBV replication was established in C57BL/6 mice by infection with AAV-HBV encoding for a wild-type (WT) or AAV-HBV-e(−) encoding for an HBeAg(−) HBV variant (E(−)). Analyses of CD8 T cell responses were performed at week 5 after the start of *TherVacB*, i.e., one week after MVA-boost vaccination. (**A**) Frequencies of CD8 T cells directly detected ex vivo in splenic and hepatic lymphocyte fractions by flow-cytometry. (**B**,**C**,**E**) HBV-core-specific CD8 T cells detected by flow-cytometry directly ex vivo through staining with C_93_ multimer: (**B**) exemplary flow cytometry results and (**C**) overall frequencies of hepatic core-specific CD8 T cells; (**E**) frequencies of splenic core-specific CD8 T cells (*n* ≥ 3 mice per group). (**D**,**F**) Frequencies and expression levels of PD-1 on (**D**) liver-associated and (**F**) on splenic, HBV-core-specific CD8 T cells. (**A**,**C**–**F**) Symbols represent individual mice. Statistical analyses were performed using (**A**,**C**,**E**) the Kruskal–Wallis test with Dunn’s multiple comparison correction and (**D**,**F**) the Mann–Whitney test. Asterisks mark statistically significant differences: * *p* < 0.05; *ns*—not significant.

**Figure 5 vaccines-09-00841-f005:**
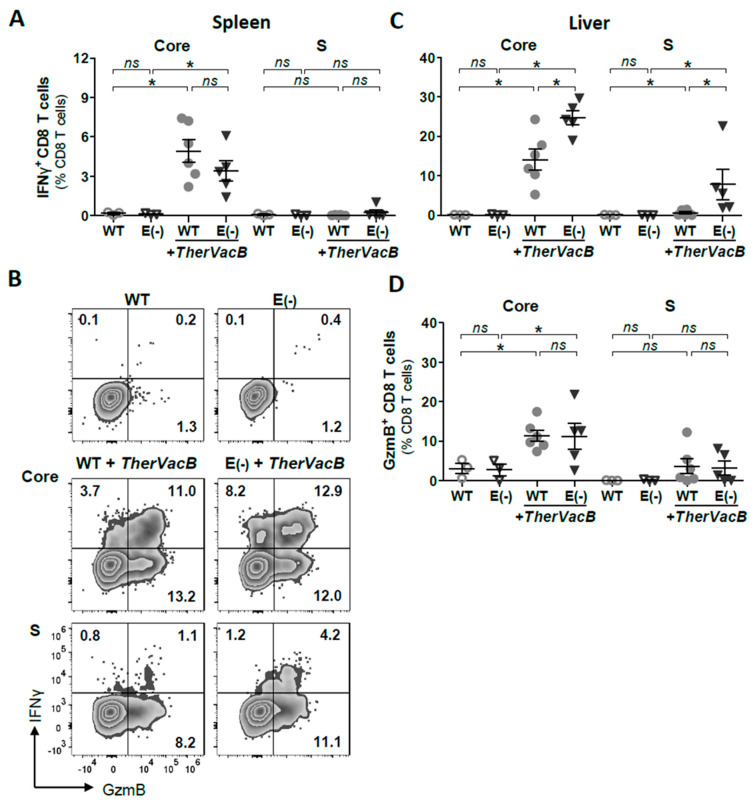
Functionality of HBV-specific CD8 T cells after therapeutic vaccination of mice replicating HBeAg(+) and HBeAg(−) HBV. Persistent HBV replication was established in C57BL/6 mice by infection with AAV-HBV encoding for a wild-type (WT) or AAV-HBV-e(−) encoding for an HBeAg(−)HBV variant (E(−)). Analyses of CD8 T cell responses were performed at week 5 after the start of *TherVacB*, i.e., one week after MVA-boost vaccination. Frequencies of core-specific and S-specific effector CD8 T cells detected by intracellular cytokine staining after 16h ex vivo stimulation with C_93_ and S_190_ peptides: (**A**) IFNγ+ CD8 T cells in spleen; (**B**) exemplary flow cytometry results of hepatic CD8 T cells; (**C**) IFNγ+ CD8 T cells, and (**D**) GzmB+ CD8 T cells detected in the liver. (**A**,**C**,**D**) Symbols represent individual mice. Statistical analyses were performed using the Kruskal–Wallis test with Dunn’s multiple comparison correction. Asterisks mark statistically significant differences: * *p* < 0.05; *ns*—not significant.

## Data Availability

The data presented in this study are available upon appropriate request from the corresponding author. The data are not publicly available to ensure IP protection.

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
