# Peer review of "Immunogenicity and Antiviral Response of Therapeutic Hepatitis B Vaccination in a Mouse Model of HBeAg-Negative, Persistent HBV Infection"

_vaccines, 2021, doi:10.3390/vaccines9080841_

Round 1

Reviewer 1 Report

The paper by Kosinska AD et al. reports interesting data on the efficacy and safety of a therapeutic hepatitis vaccination, in a mouse model of HBeAg-negative, persistent HBV infection. The paper is well written, the methods are exhaustively described and the results are clearly reported and discussed.

Some minor comments are the following ones:

  • The authors assessed the entity of HBV replication by determining serum HBsAg and HBeAg levels but they did not tested HBV-DNA in serum. Can you explain this choice or alternatively, if possible, could you assess the level of replication after infection by quantitying serum HBV-DNA? Similarly, it would be interesting to have the kinetics of serum HBV-DNA after the treatment with the vaccine TherVacB in order to estimate the rapidity of viremia decay. Furthermore, the quantification of viremia could be also interesting in the light of its comparison with the variations in the amount of intra-hepatic HBV-DNA after treatment.
  • In MAterials and Methods section the authors should specify the numer of cycles and the lower limit of detection for the Real-Time PCR assays used for intrahepatic HBV-DNA quantification.
  • Does the animal model used in this study sustain the formation of cccDNA? If that is true, could the authors estimate the impact of the treatment with the vaccine TherVacB on the amount of cccDNA?

Overall, I suggest to accept this manuscript after minor revisions.

Author Response

The paper by Kosinska AD et al. reports interesting data on the efficacy and safety of a therapeutic hepatitis vaccination, in a mouse model of HBeAg-negative, persistent HBV infection. The paper is well written, the methods are exhaustively described and the results are clearly reported and discussed.

We thank the reviewer for this remark on the quality of the manuscript.

Some minor comments are the following ones:

  • The authors assessed the entity of HBV replication by determining serum HBsAg and HBeAg levels but they did not test HBV-DNA in serum. Can you explain this choice or alternatively, if possible, could you assess the level of replication after infection by quantifying serum HBV-DNA? Similarly, it would be interesting to have the kinetics of serum HBV-DNA after the treatment with the vaccine TherVacB in order to estimate the rapidity of viremia decay. Furthermore, the quantification of viremia could be also interesting in the light of its comparison with the variations in the amount of intra-hepatic HBV-DNA after treatment.

We chose to determine HBsAg and HBeAg in serum in a quantitative fashion as this is possible with the amount of serum that can be obtained from a mouse through repeated bleeding. Unfortunately, we cannot provide the kinetics of serum HBV DNA during the treatment due to the limited material. We obtain approximately 30-40 µl serum per bleeding time point which is used to perform all serological and virological analyses. At east 50 µl would be needed for a reliable extraction of DNA and quantification of HBV viremia. Thus, in our hands, quantification of HBsAg and HBeAg is more reliable, stable and sensitive than HBV PCR in serum in low titer HBV infection.

We fully agree that intrahepatic DNA only has a limited value because PCR cannot distinguish between HBV replication and the AAV-HBV replication template – and cannot determine whether the AAV-HBV genome reached the hepatocyte nucleus to allow transcription of HBV RNAs. We have therefore now included the quantification of HBV DNA in serum of the mice at the end time point (now Fig. 3C), in which we are able to obtain enough serum for analyses (100µl). For comparison below we present the HBV DNA quantification in 100µl serum and HBsAg level detected in 10µl serum at week 5. In both cases vaccinated mice show negative results.

As expected, there is the weak correlation between intrahepatic and serum DNA when both are correlated. Positive signal for intrahepatic HBV DNA despite negative viremia is most likely due to remaining AAV-HBV DNA. This explanation was added in the text.

  • In Materials and Methods section the authors should specify the number of cycles and the lower limit of detection for the Real-Time PCR assays used for intrahepatic HBV-DNA quantification.

We now provide the additional information about HBV PCRs for the reader in Materials & Methods section. The lower limit of detection is approximately 10-100 HBV copies per 100 ng liver tissue.

  • Does the animal model used in this study sustain the formation of cccDNA? If that is true, could the authors estimate the impact of the treatment with the vaccine TherVacB on the amount of cccDNA?

The AAV-HBV model used in this study does support the formation of cccDNA. In contrast to natural infection, however, the formation of cccDNA results from a recombination event within the AAV-HBV genome and not from intracellular recycling of capsids to the nucleus (Ko C, et al., Antiviral Research, in press). Therefore, “cccDNA” is lost over time even without any treatment.

Overall, I suggest to accept this manuscript after minor revisions.

Reviewer 2 Report

In this study, Kosinska et al. aimed to determine the impact of HBeAg status on HBV-specific immune responses induced by therapeutic TherVacB vaccination. Although with an imperfect model, as pointed out by the authors that it only models the loss of HBeAg expression due to mutation but not due to immune activation, this study presents interesting results that improve the understanding of how HBeAg status affects immune responses. However, the following concerns should be addressed for the publication.

Major concerns:

  1. The study is based on the results acquired 1 week after TherVacB vaccination. Is the immune control achieved by TherVacB vaccination long lasting? Have authors look further after vaccination, such as 1 month after vaccination? Do authors see similar functional and phenotype differences in CD8 T cells 1 month after vaccination?

  1. A lower dose of AAV-HBV (1x1010) was used to achieve low viremia and relatively low-level antigen expression to facilitate the induction of stronger effector CD8 T cell response. Have authors used the same dose as Figure 1 (3x1010)? Can immune control still be achieved? This would help to expand the application of TherVacB to broader patients.

Minor concerns:

  1. Please specify how many times each experiment was independently performed.

  1. Figure 4 legend requires editing.

Author Response

In this study, Kosinska et al. aimed to determine the impact of HBeAg status on HBV-specific immune responses induced by therapeutic TherVacB vaccination. Although with an imperfect model, as pointed out by the authors that it only models the loss of HBeAg expression due to mutation but not due to immune activation, this study presents interesting results that improve the understanding of how HBeAg status affects immune responses. However, the following concerns should be addressed for the publication.

We thank the reviewer for this remark on the quality of the manuscript. However, due to the short time available for revision of the manuscript, no additional in vivo experiments could be performed. This would require at least 4 months.

Major concerns:

  1. The study is based on the results acquired 1 week after TherVacB vaccination. Is the immune control achieved by TherVacB vaccination long lasting? Have authors look further after vaccination, such as 1 month after vaccination? Do authors see similar functional and phenotype differences in CD8 T cells 1 month after vaccination?

Development of a memory immune response and observing the long-term effects of therapeutic vaccination certainly are very interesting aspects. In this study we only monitored the mice up to 1 week after vaccination. At this time point mice receiving TherVacB vaccination showed undetectable HBsAg, HBeAg, serum HBV-DNA (we included these data in a new Figure 3C) and barely detectable intrahepatic HBV DNA. We did not perform a long-term experiment.

However, according to our experience and previously published data, TherVacB induces a memory response and a reduction in HBV parameters shortly after TherVacB as observed here is a strong indicator of long-term immune control and elimination of HBV (Kosinska et al., Sci Rep 2019; Michler and Kosinska et al., Gastroenterology 2020).

As for the differences in T-cell phenotype between mice replicating HBeAg(+) and HBeAg(-) HBV – based on our studies and those reported by others (Michler and Kosinska et al., Gastroenterology 2020, Pallett et al. JEM 2018) - we believe that PD-1 expression together with high cytokine production is predictive for the formation of liver-resident immune memory population in both HBeAg(+) and HBeAg(-) mice. The PD-1 levels detected in this study for HBeAg(+) mice by no means indicate progression towards an exhausted phenotype as these levels are much too low compared to our previous study (Michler and Kosinska et al., Gastroenterology 2020).

A lower dose of AAV-HBV (1x1010) was used to achieve low viremia and relatively low-level antigen expression to facilitate the induction of stronger effector CD8 T cell response. Have authors used the same dose as Figure 1 (3x1010)? Can immune control still be achieved? This would help to expand the application of TherVacB to broader patients.

We agree with the reviewer that using high dose AAV-HBV may more closely resemble the situation in patients, in particular those with HBeAg-mutant HBV. However, infection with the dose of 3x1010 resulted in HBV replication and antigen expression levels at which mice do not respond to TherVacB vaccination without downregulating HBV expression levels by siRNA treatment (Michler and Kosinska et al., Gastroenterology 2020). In order to avoid a bias by the addition of siRNA treatment, we decided to rather lower the AAV-HBV dose.

However, we have shown in our study that the course of HBeAg(+) and HBeAg(-) HBV infection does not differ between high and low titer mice. Moreover, only minor differences were observed for the TherVacB-mediated immune and therapeutic effects in mice with HBeAg(+) and HBeAg(-) HBV replication. We do not expect any change if the infection level would increase. One may postulate that the situation may be different if the HBeAg(-) infection per se would be accompanied by an immune activation, but that is not the case in our model.

Minor concerns:

  1. Please specify how many times each experiment was independently performed.

The experiments presented in this manuscript were performed once due to the strict ethical reasons under the local animal experimentation law (3R rules). This information is now provided for the reader in Materials & Methods section

  1. Figure 4 legend requires editing.

We thank reviewer for noticing the mistake. We have corrected the Figure 4 legend.

Round 2

Reviewer 2 Report

The "Reduction" in 3R rules recommends that you minimize the number of animals per experiment or study. It does not justify that you only perform experiment once. Reproducibility is critical in scientific research. But I appreciate that you provide the information in the Materials and Methods section.